# Machine Reading Comprehension using Case-based Reasoning

**Dung Thai[1], Dhruv Agarwal[1], Mudit Chaudhary[1], Wenlong Zhao[1], Rajarshi Das[2*]**
**Manzil Zaheer[2], Jay-Yoon Lee[4], Hannaneh Hajishirzi[5], Andrew McCallum[1]**
[1]University of Massachusetts Amherst
[2]AWS AI Labs, [3]Google Research, [4]Seoul National University, [5]University of Washington
{dthai,dagarwal,mchaudhary,wenlongzhao,mccallum}@cs.umass.edu

## Abstract

We present an accurate and interpretable method for answer extraction in machine reading comprehension that is reminiscent of case-based reasoning (CBR) from classical AI. Our method (CBR-MRC) builds upon the hypothesis that contextualized answers to similar questions share semantic similarities with each other. Given a test question, CBR-MRC first retrieves a set of similar cases from a non-parametric memory and then predicts an answer by selecting the span in the test context that is most similar to the contextualized representations of answers in the retrieved cases. The semi-parametric nature of our approach allows it to attribute a prediction to the specific set of evidence cases, making it a desirable choice for building reliable and debuggable QA systems. We show that CBR-MRC provides high accuracy comparable with large reader models and outperforms baselines by 11.5 and 8.4 EM on NaturalQuestions and NewsQA, respectively. Further, we demonstrate the ability of CBR-MRC in identifying not just the correct answer tokens but also the span with the most relevant supporting evidence. Lastly, we observe that contexts for certain question types show higher lexical diversity than others and find that CBR-MRC is robust to these variations while performance using fully-parametric methods drops. [1]

## 1 Introduction

Machine reading comprehension (MRC) aims to measure the ability of models to understand and reason over a specified sequence of text. In the extractive setting, the task requires a model to answer a question by reading a set of one or more passages, referred to as the context, and identifying a span of text from that context as the answer.

---
*Work done while at University of Washington.
[1]Our code is available at https://github.com/dungtn/cbr-txt

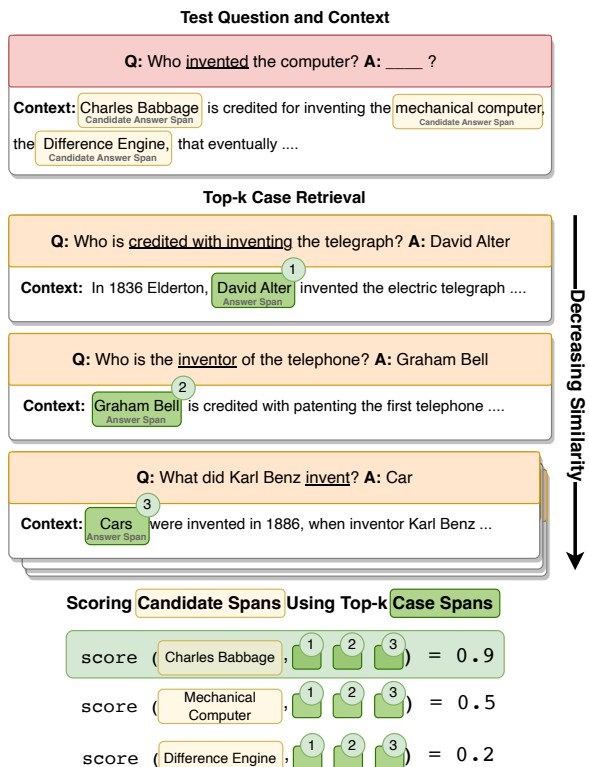

Figure 1: **Overview of CBR-MRC.** *Top:* given a test question, CBR-MRC first retrieves similar cases from a memory of known (question, context, answer) triples. *Bottom:* CBR-MRC scores each candidate answer span by comparing its contextualized representation with the answer spans of the retrieved cases to output a score. The candidate with the highest aggregate score with the answer spans from the cases is output as the prediction ("Charles Babbage").

Current state-of-the-art machine readers (Devlin et al., 2019; Choi et al., 2018; Liu et al., 2019; Lewis et al., 2020; Izacard and Grave, 2021) use fine-tuned transformer-based models (Vaswani et al., 2017). However, the prediction mechanisms in these models are largely opaque to humans, limiting their interpretability and maintainability in real-world scenarios (Ribeiro et al., 2016; Yang et al., 2018; Camburu et al., 2018). This becomes particularly relevant when practitioners need to determine

the cause of erroneous predictions, for instance, to patch models deployed in production systems. Moreover, to build reliable QA systems, it is essential for models to not only predict the correct answer but also provide evidence to support the predictions (Thayaparan et al., 2020; Zeng et al., 2020).

To address these limitations, we introduce **CBR-MRC** — a semi-parametric approach for MRC that makes predictions by explicitly reasoning over a set of retrieved answers for similar questions and their contexts from a non-parametric memory. Our key idea is that contextualized answers to similar questions share representational similarities. We posit that a (question, passage, answer) triple captures a set of *latent relations*, which can be understood to represent the *types* or *intents* present within a question given the accompanying context. We use the intersection of latent relation sets between questions to learn a similarity function that can then be used to predict answers to questions with similar intents. More concretely, our approach first compares the contextualized representations of candidate answer spans in the target context with gold answer spans of similar questions from memory. Then, the candidate with the highest aggregate similarity with the gold case answer spans is selected as the predicted answer. For example, in Figure 1, the question "*Who invented the computer?*" shares the same latent relations with the question "*Who is the inventor of the telephone?*", and the contextualized span similarity with the case answer "*Graham Bell*" is maximized when compared with the correct answer "*Charles Babbage*" and low when compared with other candidate spans such as "*mechanical computer*".

CBR-MRC adapts the case-based reasoning (Schank, 1983) paradigm from classical AI (Kolodner, 1983; Rissland, 1983; Leake, 1996), which typically consists of four steps — retrieve, reuse, revise, and retain. CBR-MRC uses the first two — **retrieval** from a memory of cases, or *casebase*, using the embedding of the test question followed by **reusing** reasoning patterns encoded in the contextualized answer embeddings of the retrieved cases to select the most similar span from the test context as the predicted answer. To the best of our knowledge, we are the first to propose a novel adaptation of the CBR framework for the task of question answering over unstructured text. We empirically verify CBR-MRC on NaturalQuestions and NewsQA, two

common MRC datasets, and find our method outperforms strong baselines by up to 11.53 EM. The inference procedure of CBR-MRC is not just accurate but also interpretable — we know which cases the model uses for its prediction and how much each case contributes. Further, the semi-parametric nature of CBR-MRC allows for efficient and accurate domain adaptation by simply adding new cases to the casebase without the need to modify any parameters.[2] Lastly, we also find that our method shows higher gains compared to fully-parametric models when the training set contains lexically diverse contexts for questions with similar latent relations, demonstrating the robustness of CBR-MRC when questions may be answered using several different passages.

In summary, our contributions are as follows:

1. We propose CBR-MRC — a novel semi-parametric approach for MRC that outperforms strong baselines on multiple datasets and settings by up to 11.53 EM on NaturalQuestions and 8.4 EM on NewsQA.

2. We show that CBR-MRC makes predictions by identifying the correct supporting evidence. When presented with gold and noisy evidence sentences, CBR-MRC predicts the correct answer span with 83.2 Span-F1 and 74.6 Span-EM compared to BLANC (Seonwoo et al., 2020a) with 79.0 Span-F1 and 66.8 Span-EM on NaturalQuestions.

3. We show the ability of CBR-MRC for efficient and accurate few-shot domain adaptation. Compared to a strong baseline (Friedman et al., 2021), we achieve a 5.5 EM gain on RelEx (Levy et al., 2017) and at-par performance on BioASQ (Tsatsaronis et al., 2015).

4. We show that questions with different sets of latent relations can be expressed with varying degrees of lexical diversity in their accompanying contexts and analyze model behavior on such questions. As lexical diversity increases, fully-parametric methods drop by up to 15.4 F1, while CBR-MRC shows robust performance with drops of only up to 4.1 F1.

---

[2]Case augmentation can be extended to make point-fixes for specific inference errors without expensive re-training.

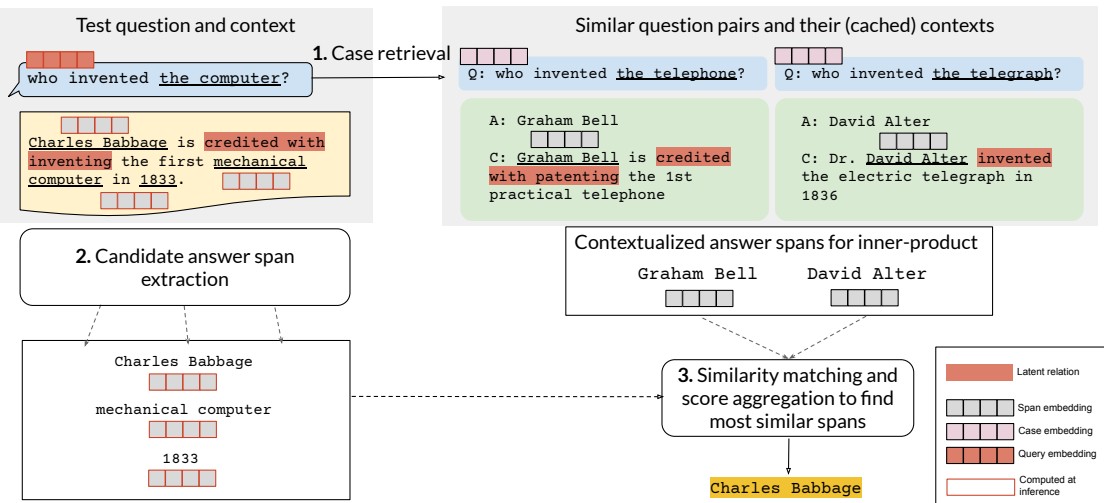

Figure 2: **Inference with CBR-MRC.** For a given target query, (1) other similar queries (and their contexts) are retrieved; (2) candidate answer spans are extracted from the target context; (3) the candidate spans in the target context are ranked w.r.t. answer spans of the retrieved queries by comparing their contextualized embeddings. Finally, the span with the highest inner-product similarity is selected as the prediction. Note that the casebase questions and their corresponding answer span embeddings are pre-computed and cached.

## 2 Related Work

**Machine Reading Comprehension.** MRC evaluates machine ability to reason over natural language by training it to answer questions on a given passage. Several variations of this task exist, such as cloze-style (Hermann et al., 2015; Yagcioglu et al., 2018), multiple-choice (Richardson et al., 2013; Lai et al., 2017), extractive (Yang et al., 2015; Trischler et al., 2016), and generative question-answering (Bajaj et al., 2016; Kočiský et al., 2017). Our work focuses on the extractive MRC setting, which aims to identify the correct answer span within a passage given a test question.

Interpretable methods for MRC have also seen much recent interest. Thayaparan et al. (2020) highlights the challenges facing MRC in terms of explainability and its impact on MRC performance. Several methods (Yu et al., 2014; Min et al., 2018; Gravina et al., 2018) have attempted to improve interpretability in MRC by identifying the supporting sentence for the answer. However, these methods rely on the availability of gold-supporting sentences, which may not always be present. Other methods utilize graph networks (Ye et al., 2019; Tu et al., 2019) and attention mechanisms to capture the attended parts of text or relations (Seo et al., 2016; Shao et al., 2020). In this work, we improve interpretability by providing a novel inference procedure that allows us to clearly attribute predictions to previously seen instances that were used in the decision-making process.

**Case-based Reasoning.** Previous work (Das et al., 2020a,b, 2022; Thai et al., 2022; Yang et al., 2023) has demonstrated strong performance using CBR on knowledge graph reasoning tasks. CBR has also been successfully applied to the task of semantic parsing (Das et al., 2021; Pasupat et al., 2021; Awasthi et al., 2023), where methods typically retrieve the logical forms of similar cases and pass them through a seq2seq model as input along with a target query. Both settings, however, operate over structured spaces, where similarities between instances can easily be determined by collecting and comparing symbolic patterns in the data, e.g., paths between nodes in a KG, or syntactic similarities in semantic parsing. In this work, however, we adapt CBR to operate over unstructured text, and it is not obvious how previous approaches can be re-used to work in this setting out-of-the-box.

A concurrent work (Iyer et al., 2023) performs answer sentence selection by using Graph Neural Networks to model interactions among questions and answers, assuming that for each question-answer, there exist similar question-answers in the training data. We instead focus on the more fine-grained task of answer span prediction from contexts and use transformer models to score question-question and answer-answer similarities between each retrieved case and a test example.

**In-Context Learning (ICL).** ICL is an emergent property observed in large language models (LLMs) (Brown et al., 2020; Min et al.,

2022) showing remarkable success in several NLP tasks (Cheng et al., 2022; Lewkowycz et al., 2022; Dunn et al., 2022). To make predictions, a model is simply prompted with demonstrations for a target task and a test instance. Recent work (Rubin et al., 2021; Liu et al., 2021; Li et al., 2023; Khattab et al., 2022) has shown strong performance by retrieving relevant demonstrations. This setting can be understood as a specific instantiation of the case-based reasoning framework, where the "reuse" step uses ICL instead of a fine-tuned model. ICL, however, is sensitive to the prompt (Chen et al., 2022), leading to black-boxed inference and reducing reliability of predictions. Problem decomposition and iterative prompting methods, such as chain-of-thought (Wei et al., 2022) and least-to-most (Zhou et al., 2022; Drozdov et al., 2022), attempt to address this issue by converting complex problems into intermediate steps. In contrast to methods reliant on emergent pre-training behavior, our work *explicitly* encodes the relational similarity hypothesis in a trained model to reuse exemplars for making predictions.

## 3 Method

We now formally describe CBR-MRC (Figure 2). A *case* in CBR consists of a problem and its solution. Formally, we define a case $c$ as a tuple $\{q, \mathcal{A}, p\}$, representing a question, its corresponding answer set[3], and the passage context that contains each element in the answer set. The context $p$ is a span of text that not only contains all $a \in \mathcal{A}$ but also captures one or more latent relations between the question and the answer. We refer to a collection of all cases as the *casebase* $\mathcal{B}$, which is typically set to the full training data and may continually be augmented to incorporate new data patterns. In the MRC setting, annotations for $p$ for all $c \in \mathcal{B}$ are assumed to be available.

### 3.1 Case Retrieval

Given an input question $q$, CBR-MRC retrieves *similar* cases from the casebase, where similarity is determined by the overlap of latent relations encoded by the query and the casebase questions. Formally, let $\mathcal{R}_q$ be the set of latent relations expressed in $q$. For example, for the question "*who invented the telephone?*", the latent relation expressed is the concept "*invented by*".[4] A case $c_k := \{q_k, \mathcal{A}_k, p_k\}$

___

[3]$\mathcal{A}$ is a singleton if every question has a unique answer.

[4]Note that CBR-MRC does not require explicit annotations of question relation types, e.g., mapping to KB schema

is similar to $q$, if the latent relations expressed in $q$ and $q_k$ overlap, i.e., $\mathcal{R}_q \cap \mathcal{R}_{q_k} \neq \emptyset$. For the above example, a similar question may be "*Who was the inventor of the television?*". We hypothesize that these latent relations are captured in dense representations obtained from language models; thus, relational overlap can be assessed via vector similarity scores. In our experiments, we use a pre-trained BERT (Devlin et al., 2019) model to encode questions and use the representation of the [CLS] token for similarity calculations.

To ensure that a question representation faithfully captures the expressed relations and is not influenced by the entities in the question, we replace mentions of all entities with a [MASK] token, *e.g.,* "*who invented the telephone?*" becomes "*who invented the* [MASK]*?*". Previous work (Soares et al., 2019) has shown this technique to be useful in learning entity-independent relational representations. In CBR-MRC, masking removes spurious similarities, where the presence of a common entity results in two relationally divergent questions being adjudged as similar, for e.g., "*who invented the telephone?*" and "*what is the telephone used for?*" should be considered dissimilar in our setup. In practice, we use T-NER (Ushio and Camacho-Collados, 2021), an automatic entity recognizer, to identify mentions of all entities in the question.

The similarity score between two masked queries is computed as the inner product between their normalized vector representations (cosine similarity). The representations for all questions in the casebase are pre-computed and cached. During inference, we retrieve the top-$k$ similar cases for a query by performing a fast nearest-neighbor search over the cached representations.

**Case retrieval during training.** To minimize the number of retrieved cases with a low overlap in latent relations during training, we additionally employ two filtering heuristics — (1) we use a minimum threshold of 0.95 for the cosine similarity score to filter out dissimilar cases, and (2) we use a *wh-filter* to remove retrieved cases where the question keyword (e.g., *"who"*, *"where"*, *"when"*) present in the case question does not match the keyword present in the input query.

### 3.2 Case Reuse

We hypothesize that contexts associated with similar questions also share relational similarities. For

___

relations. It instead leverages such semantics implicitly.

instance, Figure 2 shows the context surrounding the answer span of the retrieved case "***Graham Bell is credited with patenting...***" and the target context surrounding the gold answer "***Charles Babbage is credited with inventing...***" In CBR-MRC, we use the contextualized embeddings of the answer spans from retrieved cases to find this semantic overlap within the target context. For each candidate answer span in the context, a similarity score is explicitly computed with answer representations from the retrieved cases.

For efficient processing, we employ a dense retriever that uses contextualized answer spans from past cases as queries and searches for the target answer over the input passage embeddings. Given an input question $q$ and an input passage $p$, we search for a text span $s \in \mathcal{S}_p$, the set of all possible spans in $p$, whose representation is the most similar to representations of answer spans in similar question contexts. We employ a standard BERT-base model as the passage encoder of the dense retriever. Due to the combinatorial complexity of considering all $s \in \mathcal{S}_p$, we follow Seo et al. (2019) and consider only a subset of candidate spans in $p$ likely to be the answer. In particular, we consider a set of candidates $\mathcal{S}_{\text{cand}} \subseteq \mathcal{S}_p$ such that $\mathcal{S}_{\text{cand}}$ is composed of only entity mentions, date-time strings, numbers, and quoted strings that can be extracted from $p$. To increase span coverage, we include all n-grams up to length three. See A.1 for more details regarding the candidate span extraction process.

We now formulate the task to predict the answer $\hat{a}$ to the question $q$ from a context passage $p$ given its relevant past cases. Let $c_k := \{q_k, \mathcal{A}_k, p_k\}$ be the $k$-th retrieved case for the given question $q$. For an answer $a_k \in \mathcal{A}_k$, let $\text{Enc}(a_k)$ be the contextualized answer span embedding pooled from the embeddings of the answer tokens in $p_k$. Similarly, for all candidate answers $s \in \mathcal{S}_{\text{cand}}$, let $\text{Enc}(s)$ be their contextualized embeddings pooled from the encoded tokens in context $p$. The predicted answer span based on the $k$-th retrieved case is given by

$$\hat{a}^{(k)} = \arg\max_{s \in \mathcal{S}_{\text{cand}}} \max_{a_k \in \mathcal{A}_k} \text{Enc}(s)^T \text{Enc}(a_k),$$

i.e., the candidate answer span most similar to the answer set $\mathcal{A}_k$ based on similarity scores computed using inner products. At inference, the similarity scores of each candidate span are aggregated across the top-$k$ retrieved cases by taking the maximum scores. The candidate span with the highest aggregated score is predicted as the answer.

### 3.2.1 Training

The passage encoder must be fine-tuned to maximize similarity scores between answer span embeddings of similar questions. We fine-tune a pre-trained BERT model with a contrastive loss (Khosla et al., 2020) function using answer and non-answer spans.

Specifically, consider a train question $q_t$, its set of answers $\mathcal{A}_t$, and its set of candidate spans $\mathcal{S}_{\text{cand},t} \subseteq \mathcal{S}_{p_t}$ derived from the passage $p_t$. We use the gold answer set $\mathcal{A}_t$ as the set of positive candidate spans, and define the set of negative candidate spans as $\mathcal{S}_{\text{cand},t} \setminus \mathcal{A}_t$. Let $\text{sim}(\cdot, \cdot) := \text{Enc}(\cdot)^T \text{Enc}(\cdot)$ be the similarity function (dot product or cosine similarity) between vector representations. For the top-$k$ similar questions as described in the previous section, we fine-tune the passage encoder with the following objective derived from the soft nearest-neighbor loss (Salakhutdinov and Hinton, 2007; Frosst et al., 2019):

$$\mathcal{L}_t = -\log \sum_{s \in \mathcal{A}_t} \max_k \max_{a_k \in \mathcal{A}_k} \exp(\text{sim}(a_k, s)/\tau)$$
$$+ \log \sum_{s \in \mathcal{S}_{\text{cand},t} \setminus \mathcal{A}_t} \max_k \max_{a_k \in \mathcal{A}_k} \exp(\text{sim}(a_k, s)/\tau),$$

where $\tau$ is a temperature hyperparameter. The loss for the entire training set is simply the mean of $\mathcal{L}_t$ over all $q_t \in \mathcal{Q}$, the training set. Among the top-$k$ retrieved cases, we pick the spans with the highest similarity scores. This loss function encourages the similarity scores of the gold answer spans with respect to the answers in the top-$k$ cases to be higher than the similarity of other spans of texts in the input context that are not the answer.

## 4 Experiments

We evaluate CBR-MRC on two machine reading comprehension tasks — supervised answer span extraction and few-shot domain adaptation. For three MRC datasets, we use the training data to both construct the casebase as well as train our model and report performance on the in-domain test sets. We report the few-shot domain adaptation performance on two additional datasets.

### 4.1 Datasets

**NaturalQuestions (NaturalQ)** consists of over 323K questions and answers, representative of real queries people make on the Google search engine (Kwiatkowski et al., 2019). The answers are sourced from Wikipedia and annotated by

crowd workers, including both a long and short version. Our experiments utilize short answers as the gold answers and use the long answers as context. **NewsQA** has 120K crowd-sourced questions based on CNN news articles (Trischler et al., 2016). Annotation is obtained by two sets of crowd workers: one creating questions based on article highlights and the other answering them using spans from the full article. **BioASQ** is a large-scale biomedical dataset that features question-answer pairs crafted by domain experts (Tsatsaronis et al., 2015). Questions are linked to relevant scientific literature found on PubMed, with the abstracts of these articles serving as the context for MRC. **RelationExtraction (RelEx)** comprises question-answer pairs that express Wikidata relations between entities mentioned in Wikipedia articles (Levy et al., 2017). The dataset is derived from the WikiReading (Hewlett et al., 2016) slot-filling dataset through predefined templates.

We utilize the curated version of all the previously mentioned datasets from the MRQA shared task (Fisch et al., 2019), including annotations for the gold answer spans within their context. The gold context for constructing a case, as described in Section 3, is a short paragraph surrounding the gold answer span taken from the originally provided text. Additionally, note that we use BioASQ and RelEx to *only evaluate* in the few-shot setting and not train our models.

## 4.2 Experiment Setup

We refer the reader to Appendix A.1 for more details on the models used in this work.

**Baselines.** We compare CBR-MRC to BERT (Devlin et al., 2019), RoBERTa (Liu et al., 2019), ALBERT (Lan et al., 2019), and SpanBERT (Joshi et al., 2020) as well as the SOTA baselines for our settings – BLANC (Seonwoo et al., 2020b) and MADE (Friedman et al., 2021).

**Evaluation Metrics.**[5] We report exact match (EM) and F1 scores to measure model performance. EM calculates the percentage of predictions that match all tokens of any one of the correct answers for a given question. F1 score measures the token overlap between the prediction and the correct answer. As these metrics do not take into account the

---

[5] (1) All reported numbers are from single runs on the test set unless noted otherwise. (2) Hyperparameters used by our models were tuned on a development set and the details are included in the appendix.

| Model | NewsQA | | NaturalQ | |
|---|---|---|---|---|
| | EM | F1 | EM | F1 |
| BERT | 50.11 | 65.07 | 64.48 | 76.39 |
| ALBERT | 51.18 | 66.02 | 63.81 | 75.89 |
| RoBERTa | 52.36 | 67.28 | 66.33 | 78.54 |
| MADE | 56.55 | **72.12** | 68.86 | 80.09 |
| SpanBERT | 52.85 | 67.93 | 66.60 | 78.31 |
| BLANC | 55.52 | 70.31 | 68.33 | 80.04 |
| SpanBERT$_{\text{LARGE}}$ | 53.84 | 69.06 | 69.14 | 80.66 |
| BLANC$_{\text{LARGE}}$ | 57.40 | 72.36 | 70.59 | 81.99 |
| **CBR-MRC** | **64.95** | 69.17 | **82.12** | **83.10** |

Table 1: **MRC performance on NewsQA and NaturalQ.** CBR-MRC outperforms all baselines in finding the exact match of the answers. On NewsQA, our model shows +8.4 EM over MADE but has lower F1 which indicates CBR-MRC focuses on getting the right answer rather than overfitting to the answer tokens.

surrounding context of the answer span (i.e., the supporting evidence), we also report Span-EM and Span-F1, following Seonwoo et al. (2020b), which consider overlapping indices between the predicted and gold answer spans and exclude answers with irrelevant supporting text.

## 4.3 Results

We report our main MRC results on NewsQA and NaturalQ in Table 1. We find that CBR-MRC consistently outperforms all baselines. On NaturalQ, CBR-MRC achieves the state-of-the-art, outperforming the next best model (BLANC$_{\text{LARGE}}$) by 11.53 EM. Similarly, on NewsQA, CBR-MRC shows an improvement of 12.56 EM over BLANC$_{\text{LARGE}}$, and 8.4 EM over the next best model (MADE). Note that the best baseline model (BLANC$_{\text{LARGE}}$) has three times the number of parameters as CBR-MRC. Furthermore, our model shows performance gains in EM but a lower F1 which indicates CBR-MRC focuses on getting the correct answer rather than over-fitting to the answer tokens.

### 4.3.1 Supporting Evidence Identification

It is becoming increasingly important to train models to predict the *right* answer for the *right* reasons. Since CBR-MRC identifies the answer span by comparing it with multiple answer-containing contexts of similar cases, we hypothesize that it can cut through the noise and identify the correct occurrence of the span that *exactly* answers the question. Our main baseline is BLANC (Seonwoo et al., 2020b), which trains an auxiliary model specifically for context prediction.

| Dataset | Model | Span-EM | Span-F1 |
|---|---|---|---|
| NaturalQ | BERT | 60.63 | 72.92 |
| | ALBERT | 60.31 | 72.66 |
| | RoBERTa | 62.59 | 75.07 |
| | SpanBERT | 62.71 | 75.16 |
| | BLANC | 64.57 | 76.99 |
| | SpanBERT$_{LARGE}$ | 65.28 | 77.62 |
| | BLANC$_{LARGE}$ | 66.75 | 79.04 |
| | **CBR-MRC** | **74.64** | **83.16** |
| NewsQA | BERT | 45.53 | 59.18 |
| | ALBERT | 46.54 | 60.12 |
| | RoBERTa | 47.43 | 61.36 |
| | SpanBERT | 48.04 | 62.26 |
| | BLANC | 50.60 | 64.39 |
| | SpanBERT$_{LARGE}$ | 49.03 | 63.43 |
| | BLANC$_{LARGE}$ | 52.39 | 66.48 |
| | **CBR-MRC** | **54.07** | **68.21** |
| NaturalQ* | RoBERTa | 60.12 | 65.99 |
| | SpanBERT | 57.63 | 63.47 |
| | BLANC | 61.43 | 67.07 |
| | **CBR-MRC** | **71.30** | **81.04** |

Table 2: **Identifying the correct supporting evidence.**
We report Span-F1 and Span-EM scores for CBR-MRC
and baseline models, indicating how often the span with
the correct supporting evidence is identified versus se-
lecting either the incorrect answer or a spurious occur-
rence of the correct answer. We report baseline numbers
for BLANC as reported in their paper. NaturalQ* is a
subset of NaturalQuestions containing questions with
answers having at least two mentions within the context.

Table 2 highlights the performance gains of CBR-
MRC compared to BLANC and several strong base-
lines. Our method shows improvements on Natu-
ralQ by 7.89 Span-EM and 4.12 Span-F1 points
in predicting the span indices. On NewsQA, it im-
proves performance by 1.68 Span-EM and 1.73
Span-F1 points compared to BLANC. To deter-
mine the gains due to predicting the answer span
with the correct supporting evidence, we follow
the approach in BLANC and evaluate performance
on the NaturalQuestions (NaturalQ*) subset that
contains questions with answers that have at least
two mentions within the context. Our method out-
performs BLANC on this subset by 6.73 and 4.05
points for Span-EM and Span-F1, respectively.

### 4.3.2 Robustness to Lexical Diversity

CBR-MRC performs answer extraction by explic-
itly comparing span similarities of the target con-
text with the context spans in the filtered casebase.
Purely parametric methods can also be seen as fol-
lowing a similar mechanism, albeit performing the

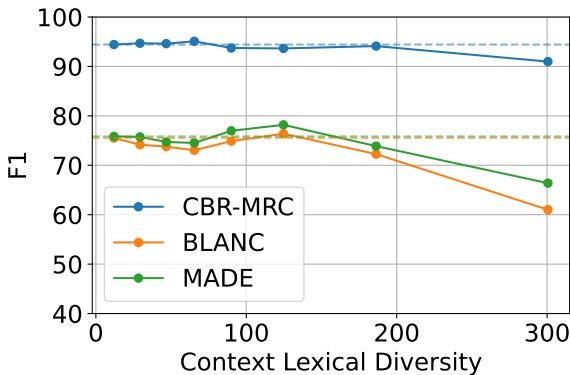

Figure 3: **Robustness to lexical diversity in passages.**
Lexical diversity of latent relation clusters is the num-
ber of unique tokens in passages seen at training for
that latent relation cluster (or question type). We plot
F1 performance (averaged over 6 clusterings varying
cut thresholds) on 8 buckets of increasing lexical diver-
sity for the latent relations seen at training. CBR-MRC
shows a drop in performance only up to 4.10 points,
while BLANC and MADE show drops of up to 15.38
and 11.80 points, respectively.

two steps implicitly. These models, therefore, must
rely only on the parameter values learned at train-
ing in order to handle several relational semantics,
which may each be expressed in contexts with vary-
ing levels of diversity in language form. In practice,
we find that some relations are indeed expressed
with more lexical diversity than others (Appendix
A.3). In Figure 3, we investigate how CBR-MRC
performs at inference under varying amounts of
lexical diversity observed in passage contexts at
*training*. We first cluster questions according to
their embeddings and then compute n-gram statis-
tics (Li et al., 2015; Dušek et al., 2020; Tevet and
Berant, 2020) across passage contexts found in
each cluster to measure diversity. We use Natu-
ralQuestions as a case study for our analysis and
compare performance with BLANC and MADE,
two fully-parametric baselines.

**Clustering questions by latent relations.** We
cluster questions in the training set based on the
latent relations they encode using hierarchical
agglomerative clustering (HAC) (Krishnamurthy
et al., 2012) over similarity scores. For a fair eval-
uation, we compute 6 different flat clusterings of
varying cluster tightness, i.e., how similar each
question is within a cluster. For each clustering,
we compute the per-cluster lexical diversity as the
count of the number of unique tokens in the set
of passage contexts across questions in the cluster,
normalized by the cluster size. We then cluster the

| Model | BioASQ | | RelEx | |
|---|---|---|---|---|
| | EM | F1 | EM | F1 |
| MADE$_{\text{ZERO}}$ | 43.3 | 62.9 | 72.0 | 84.9 |
| MADE$_{\text{FT}}$ | **62.5** | **74.5** | 82.3 | **90.0** |
| **CBR-MRC** | 62.4 | 64.6 | **87.8** | 88.2 |

Table 3: **Few-shot domain adaptation.** We compare CBR-MRC with two variants of the MADE adapter networks, which are also trained on NaturalQuestions. MADE$_{\text{ZERO}}$ demonstrates zero-shot performance, while MADE$_{\text{FT}}$ is further fine-tuned on 256 samples for each dataset. CBR-MRC is evaluated by simply adding the samples into the casebase without any fine-tuning.

test set by assigning each question to the cluster label of its top-1 similar train question for each of the 6 clusterings, resulting in 6 *test* clusterings (see Appendix A.4 for more details).

**Results.** Figure 3 shows the mean F1 scores between CBR-MRC and the baselines on test queries for varying values of lexical diversity observed at training for the same latent relations as the test queries. We group clusters by lexical diversity scores into buckets and report the aggregated F1 scores for each bucket.[6] On average, the performance gap between CBR-MRC and baselines tends to increase as lexical diversity increases. The min-max F1 difference, which captures the drop in performance as lexical diversity increases, is $4.10$ points for CBR-MRC while BLANC and MADE show significantly larger differences of $15.38$ and $11.80$ points, respectively. We posit that as diversity in contexts for the same latent relation increases, it is harder for fully parametric models to learn parameters that are suitable for all lexical variations. On the other hand, the semi-parametric nature of CBR-MRC allows it to partially shift this burden to test time and ground itself using only a few retrieved cases that are explicitly compared.

### 4.3.3 Few-shot Domain Adaptation

It is important for models to generalize to unseen datasets. We test CBR-MRC trained on NaturalQ on two diverse out-of-domain datasets - BioASQ for MRC (Tsatsaronis et al., 2015) and RelEx for relation extraction (Levy et al., 2017). We compare our model with the MADE adapter networks (Friedman et al., 2021) that are trained on NaturalQuestions, and further fine-tuned on 256 samples from

---

[6](1) We use $k$-means over lexical diversity scores of all clusters to construct $B = 8$ buckets. (2) We report the F1 performance on all 6 clusterings in Appendix A.5.

| Cases ($k$) | 1 | 5 | 10 | 20 |
|---|---|---|---|---|
| **EM** | 82.03 | **82.12** | 77.95 | 77.39 |
| **F1** | 82.13 | **83.10** | 78.00 | 78.45 |

Table 4: **Performance with different number of case retrievals $k \in \{1, 5, 10, 20\}$.** Both EM and F1 display the same trend. Both performance metrics increase as $k$ increases from 1 to 5. However, larger sets of retrievals ($k > 10$) reduce performance to levels worse than even $k = 1$, which indicates the introduction of noise.

the target dataset. We follow the same practice in (Friedman et al., 2021) and evaluate CBR-MRC on a held-out subset of $400$ samples. For both datasets, we use the trained model obtained from the experiment on the NaturalQuestions dataset, and the casebase for each dataset is built from the same training set that MADE also uses.

Table 3 presents the performance of CBR-MRC and MADE adapters. Our model outperforms the zero-shot adapter network (MADE$_{\text{ZERO}}$) by simply putting some samples from the target dataset into the casebase. When compared to the fine-tuned adapter (MADE$_{\text{FT}}$), our model gains 5.5 EM points on RelEx while being comparable to MADE$_{\text{FT}}$ on BioASQ. These results confirm that CBR-MRC can transfer to a new domain by simply collecting a handful of sample cases, which is significantly less expensive than model fine-tuning.

We notice a drop in F1 scores on both datasets. F1 scores evaluate the partial matching of answers. The inference procedure used by previous work, including our baselines, independently predicts the start and end spans of the answer, which encourages the inclusion of partial matches. In CBR-MRC, on the other hand, the entire candidate span is considered when making a prediction by comparing against the representations of gold spans from the casebase. This results in higher similarity scores for candidate spans that map to the same answer type, rather than matching the correct answer partially, which explains lower F1 scores for CBR-MRC.

### 4.3.4 Effect of Retrieval Quantity

We explore how the number of retrieved cases $k$ affects the reading comprehension performance, via varying $k$ on the NaturalQuestions dataset. We find that while increasing $k$ does enrich the set of contexts for referencing in the reuse step, too many retrievals may introduce irrelevant cases that degrade performance.

### 4.3.5 Effect of Modeling

We analyze CBR-MRC performance with different base models and show that CBR-MRC can improve upon them. We run the experiments on NQ with the same set of hyper-parameters.

| Model | EM | F1 | Span-EM | Span-F1 |
|---|---|---|---|---|
| BERT | 64.48 | 76.39 | 60.63 | 72.92 |
| CBR-MRC$_{BERT}$ | **82.12** | **83.1** | 74.64 | **83.16** |
| DeBERTa | 43.11 | 74.07 | 36.03 | 63.7 |
| CBR-MRC$_{DEBERTA}$ | 80.30 | 82.22 | **76.09** | 80.39 |

Table 5: **Performances of CBR-MRC variants on NQ.** We compare two variants of CBR-MRC with BERT and DeBERTa as its base model.

## 5 Conclusion

We present CBR-MRC, a semi-parametric model for machine reading comprehension that is simple, accurate, and interpretable. Our model stores a collection of cases, retrieves the most relevant cases for a given test question, and then explicitly reuses the reasoning patterns encoded in the embeddings of these cases to predict an answer. We show that our model performs well for both extracting answers and identifying supporting evidence on several MRC tasks compared to fully-parametric baselines. We also demonstrate the ability of our model to transfer to new domains with limited labeled data. Finally, we analyze our model under varying conditions of lexical diversity and find that it is robust to high lexical diversity, whereas fully-parametric models show a drop in performance.

## 6 Limitations

CBR-MRC, and the CBR framework more generally, relies on the existence of past cases to make predictions for a new problem. This can pose a challenge for composite questions, which require multi-hop reasoning, since the likelihood of enumerating each algebraic combination of reasoning patterns in the casebase is impractical. Models, thus, may only be able to match a portion of the question, resulting in partially correct reasoning. To address these limitations, future work could explore methods to explicitly encourage such compositional generalization, such as question decomposition with a recursive application of CBR.

### Ethics Statement

The objective of our research and the proposed methodology is to enhance performance on the Machine Reading Comprehension task. The datasets utilized in this study have been extensively employed in previous research and, as far as our knowledge extends, are not associated with any privacy or ethical concerns.

### Acknowledgements

We thank members of UMass IESL for helpful discussions and feedback. This work was supported in part by Amazon Digital Services, in part by IBM Research AI through the AI Horizons Network, in part by the Chan Zuckerberg Initiative under the project "Scientific Knowledge Base Construction", and in part using high-performance computing equipment obtained under a grant from the Collaborative R&D Fund managed by the Massachusetts Technology Collaborative. Any opinions, findings, and conclusions / recommendations expressed in this material are those of the authors and do not necessarily reflect those of the sponsor(s).

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

# A Appendix

## A.1 Experiments: Implementation Details

All CBR-MRC models employ the BERT-base architecture and the cased vocabulary. We use a fixed set of hyperparameters to train all our models. More specifically, we use an Adam optimizer with a learning rate of $2e-5$, epsilon $1e-8$, and max gradient norm value of $5.0$. We set the number of warm-up steps to be 8% of the training data size. All models are trained for 10 epochs, which takes 2-4 days on a single RTX-8000 GPU each, depending on the size of the dataset. The standard models we report have parameter counts of 108M (BERT, MADE, SpanBERT, BLANC), 17M (ALBERT), 124M (RoBERTa), and the large variants have a parameter count of 333M.

Our approach follows prior QA work (Seo et al., 2019) and selects a broad set of spans that may be answers to factoid-type questions, such as entity mentions, date-time strings, numbers, and quoted strings. We also include all n-grams up to three words. For date-time string extraction, we use the $datefinder$ Python package, and for entity mentions, we use the T-NER library (Ushio and Camacho-Collados, 2021).

## A.2 Experiments: Dataset Statistics

For our experiments, we use the curated versions of NaturalQuestions, NewsQA, BioASQ, and RelEx datasets from the MRQA shared task (Fisch et al., 2019). Table 6 presents the number of examples present in each of the datasets used in the paper.

| Dataset | Train | Dev |
|---|---|---|
| NaturalQ | 104,071 | 12,836 |
| NewsQA | 74,160 | 4,212 |
| BioASQ | - | 1,504 |
| RelEx | - | 2,948 |

Table 6: Dataset statistics

## A.3 Lexical Diversity Analysis: Distribution in NaturalQuestions

We evaluate the distribution of lexical diversity in the NaturalQuestions dataset. Out of the set of test questions under consideration in our case study on lexical diversity, we find that $7.46\%$, or $504$, unique (question, answer) pairs contain more than one context, which is used to extract the answer, highlighting the relevance of lexical diversity as a

consideration for models to address. A question in this set may contain up to 4 different contexts that can answer the same exact question. In Table 8, we show examples from the test set of questions that can be answered using multiple, diverse contexts.

## A.4 Lexical Diversity Analysis: Clustering questions by latent relations

We cluster questions in the training set based on the latent relations they contain using hierarchical agglomerative clustering (HAC) (Krishnamurthy et al., 2012) over masked similarity scores[7]. Importantly, since we do not have ground-truth latent relations for questions, we run our analysis on *multiple* flat clusterings, where each cluster in a clustering should represent the same semantic relations. To obtain flat clusters in a fair manner, we cut the HAC tree using $C$ thresholds, computed using $k$-means (Hartigan and Wong, 1979) over similarity scores across all questions in the training set. This yields $C$ clusterings with increasing levels of cluster tightness, i.e., how similar each question is in a cluster. We set $C$ to 6 and lower-bound the similarity scores to $0.9$ to reduce noise in the clusters. For each clustering, we then compute the per-cluster lexical diversity as the count of the number of unique tokens in the set of passage contexts across questions in the cluster, normalized by the cluster size. Finally, we cluster the test set by assigning each question to the cluster label of its top-1 similar train question for each of the $C$ clusterings, resulting in 6 test clusterings. In our analysis, we retain only unique $(q, \mathcal{A}, p)$ triples resulting in 7,213 instances from an original set of 12,836. We additionally drop 98 questions that cannot be answered by any of the three models under consideration, resulting in a final set of 7,115 test questions.

## A.5 Lexical Diversity Analysis: F1 Performance

Figure 4 shows the absolute F1 performance of CBR-MRC and the two baselines (BLANC and MADE) on each clustering separately. Note that Figure 3 is a reduction from this, in that it takes the average difference between the F1 score of CBR-MRC and the baselines across the 6 clusters.

---

[7]We restrict the number of nearest-neighbors to 20 for each question.

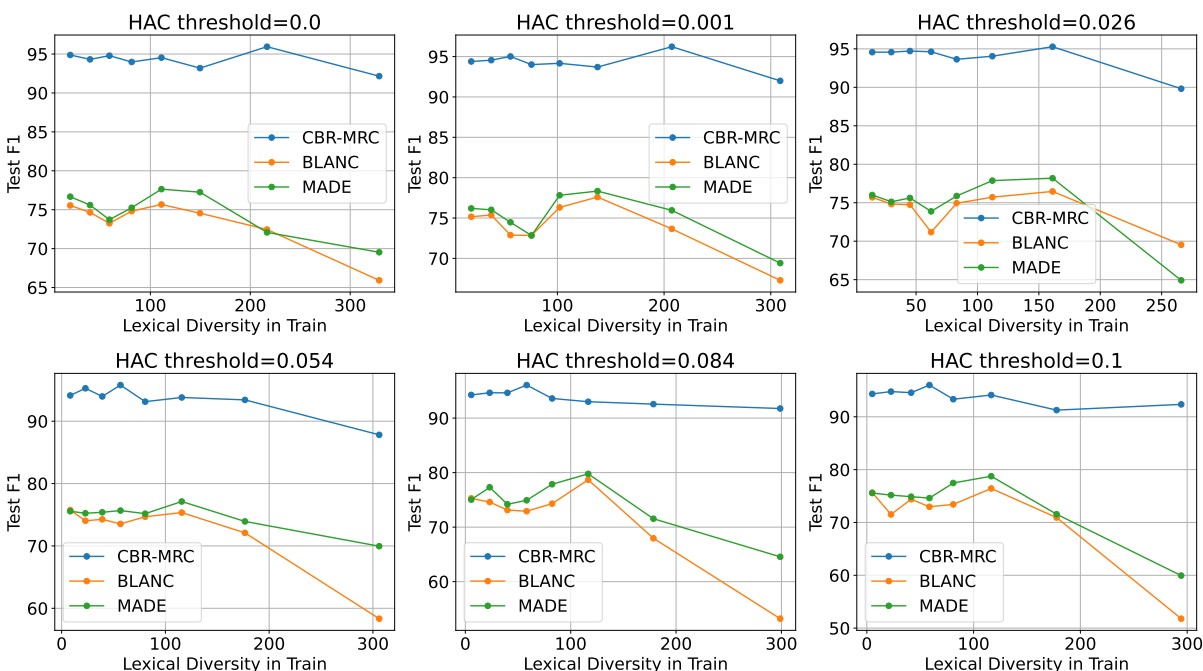

Figure 4: **F1 performance v/s lexical diversity in passage contexts on clusterings with decreasing levels of cluster-tightness.** We cluster questions by latent relations in the training set using HAC using 6 cut thresholds and assign each test question to one of these train clusters. We then bucket the clusters by lexical diversity scores and compute F1 performance of each bucket for each of the 6 clusterings shown.

## A.6 Does CBR-MRC have the correct inductive bias?

Text representations obtained from pre-trained language models have shown the ability to capture the latent semantics of text spans. In this experiment, we evaluate CBR-MRC when the parameters are set to the pre-trained BERT-base model *without* fine-tuning. As shown in Table 7, we find that our model with no fine-tuning performs surprisingly well, indicating that CBR-MRC does have the correct inductive bias. However, fine-tuning with our proposed contrastive loss leads to a significant increase in performance.

| Model | NewsQA | NaturalQ |
|---|---|---|
| BERT | 50.11 | 64.48 |
| SpanBERT | 52.85 | 66.60 |
| CBR-MRC_PRE-TRAINED | 18.63 | 46.07 |
| CBR-MRC_FINE-TUNED | 64.95 | **82.12** |

Table 7: **Pre-trained CBR-MRC performance.** We report the performance (EM) of CBR-MRC using both a pre-trained passage encoder as well as an encoder fine-tuned with the contrastive loss as described in §3. We additionally include scores from reasonable, *fine-tuned* baselines for comparison.

| Question | Answer | Context |
|---|---|---|
| when does season 5 of ruby come out | October 14, 2017 | [...] Four seasons , referred to as " Volumes " , have been released , with a fifth currently ongoing since its premiere on October 14 , 2017 . As of [...]

[...] The fifth Volume premiered on October 14 , 2017 , a date which was first announced at the RTX Austin 2017 event . The episodes [...]

[...] \<Table> \<Tr> \<Th colspan="2"> Season \</Th> \<Th colspan="2"> Episodes \</Th> \<Th colspan="2"> Originally aired \</Th> [...] \<Td colspan="1"> 5 \</Td> \<Td colspan="2"> 14 \</Td> \<Td colspan="1"> October 14 , 2017 ( 2017 - 10 - 14 ) \</Td> \<Td> [...] |
| who wrote most of the declaration of independence | Thomas Jefferson | [...] Committee of Five " to draft a declaration , consisting of John Adams of Massachusetts , Benjamin Franklin of Pennsylvania , Thomas Jefferson of Virginia , Robert R. Livingston of New York , and Roger Sherman of Connecticut . The committee [...]

[...] John Adams , a leader in pushing for independence , had persuaded the committee to select Thomas Jefferson to compose the original draft of the document , which Congress edited to produce the final version [...]

[...] The source copy used for this printing has been lost and may have been a copy in Thomas Jefferson 's hand . Jefferson 's original draft is preserved at the Library of Congress , complete with changes made by John Adams [...] |
| who plays the mom on the tv show mom | Allison Janney | [...] working as a waitress and attending Alcoholics Anonymous meetings . Her mother Bonnie Plunkett ( Allison Janney ) is also a recovering drug and alcohol addict . Christy 's daughter [...]

[...] It stars Anna Faris and Allison Janney in lead roles as dysfunctional daughter / mother duo Christy and Bonnie Plunkett [...]

[...] her relationship with her mother Bonnie [...] \<Li> Allison Janney as Bonnie Plunkett : Christy 's mother , a joyful if cynical recovering addict who is now grateful with life . She tries [...] |
| when was the us department of homeland security created | November 25 , 2002 | \<Tr> \<Th> Formed \</Th> \<Td> November 25 , 2002 ; 15 years ago ( 2002 - 11 - 25 ) \</Td> \</Tr>

[...] \<Th colspan="2"> Agency overview \</Th> \</Tr> \<Tr> \<Th> Formed \</Th> \<Td> November 25 , 2002 ; 15 years ago ( 2002 - 11 - 25 ) \</Td> [...]

[...] The Department of Homeland Security was established on November 25 , 2002 , by the Homeland Security Act of 2002 . It was intended to consolidate U.S. [...] |
| who has won the euro-vision song contest the most times | Ireland | [...] Ireland has finished first seven times , more than any other country [...]

[...] \<Table> \<Tr> \<Th> Wins \</Th> \<Th> Country \</Th> \<Th> Years \</Th> \</Tr> \<Tr> \<Td> 7 \</Td> \<Td> Ireland \</Td> \<Td> 1970 , 1980 , 1987 , 1992 , 1993 , 1994 , 1996 \</Td> \</Tr> \<Tr> \<Td> 6 \</Td> \<Td> Sweden \</Td> \<Td> 1974 , 1984 , 1991 , 1999 , 2012 , 2015 \</Td> \</Tr> \<Tr> \<Td> [...]

[...] The country with the highest number of wins is Ireland , with seven . The only person to have won more than once [...] |

Table 8: **Lexical Diversity in NaturalQuestions.** We show here unique question-answer pairs that can be answered using diverse contexts.