# OpenReview forum: "Machine Reading Comprehension using Case-based Reasoning"
_EMNLP/2023/Conference — EMNLP 2023 Findings_

### Official Review · Reviewer_Crp1 · 2023-08-04

**Soundness:** 3

**Excitement:**

3: Ambivalent: It has merits (e.g., it reports state-of-the-art results, the idea is nice), but there are key weaknesses (e.g., it describes incremental work), and it can significantly benefit from another round of revision. However, I won't object to accepting it if my co-reviewers champion it.

**Paper Topic And Main Contributions:**

This paper studies machine reading comprehension (MRC) and proposes a method called Case-based Reasoning for MRC (CBR-MRC).  The main contribution of the paper is the introduction of a semi-parametric approach for MRC that makes a prediction by explicitly reasoning over a set of retrieved answers for similar questions and their contexts from memory. The paper shows that CBR-MRC consistently outperforms all baselines on two benchmark datasets, NewsQA and NaturalQ, achieving state-of-the-art results. The paper also demonstrates that CBR-MRC can identify not just the correct answer tokens, but also the most relevant supporting span.

**Reasons To Accept:**

1) A new method for achieving high accuracy on MRC tasks.
2) The proposed method can not only achieve better accuracy but also can identify the span with the most relevant supporting evidence, which provides the use of similar cases provides interpretability.
3) The experimental results show the method's improvement on MRC datasets and the model is able to handle variations in question phrasing and still identify the correct answer.

**Reasons To Reject:**

1) The proposed system might be too complex to use in real practice. The efficiency of the case retrieval and many heuristics construction might hinder the method from being scaled up. Further scalability analysis might be needed.
2) The case retrieval may make the model focus too much on the absolute answer span position while learning the general reasoning patterns, which makes the model hard to generalize.  For example, in the main results, the model shows performance gain in EM but a lower F1 score. the author explains this by saying "The model focuses on getting the written answer rather than over-fitting to the answer tokens." However, I feel such a case retrieval will lead to more over-fitting.
3) Impact of the casebase is unclear. The method utilizes training data as the casebase for retrieval. It is unclear how that casebase influences the performance of the model. And whether the utilization of training data as the casebase would lead to over-fitting?

**Reproducibility:**

5: Could easily reproduce the results.

**Reviewer Confidence:**

4: Quite sure. I tried to check the important points carefully. It's unlikely, though conceivable, that I missed something that should affect my ratings.

---

> ### Author Rebuttal · Authors · 2023-08-29
>
> Thank you for the thoughtful review! We appreciate that you found our work novel (echoed by paQh and 2R1W), accurate, interpretable (echoed by paQh and 2R1W), and robust. Please find our detailed responses to your comments and questions below:
>
> > **Comment:** *The proposed system might be too complex to use in real practice. The efficiency of the case retrieval and many heuristics construction might hinder the method from being scaled up. Further scalability analysis might be needed.*
>
> **Response:** Our proposed approach is a sum of extremely lightweight parts. In particular, our case retrieval uses an efficient dense retriever with a small index size of around **33MB**, providing retrieval times of **only ~2.6ms**. NER for question masking and answer span extraction takes about **only ~11.8ms**. Overall, our CBR-MRC inference procedure takes **~41ms**, which is almost identical to the ~38ms latency of the baseline method, BLANC. We report all latency numbers averaged over 12,836 questions from the NQ test set.
>
> ---
>
> > **Comment:** *The case retrieval may make the model focus too much on the absolute answer span position while learning the general reasoning patterns, which makes the model hard to generalize.*
>
> **Response:** In CBR-MRC, the answer is predicted by reasoning over the contextualized representations of answer spans of similar questions. While positional information is encoded in these representations, we argue that the predominant information stored in these vectors are the semantics of the text. We, therefore, claim that **CBR-MRC does not rely on the absolute answer span positions to make predictions**. We quantitatively verify our claim by running the following analysis:
>
> For each test sample in the NQ dataset, we obtain (1) the absolute start and end token positions of CBR-MRC’s predicted answer span, and (2) the start and end token positions of the answer span from the retrieved cases (the one that has the highest similarity with a candidate span in the test context as described in L334). Then, we compute the Spearman and Pearson coefficients between the start positions of the prediction and retrieval (we repeat this calculation for the end positions). The following are the results:
>
> | | Spearman correlation | Pearson correlation|
> |:---:|:---:|:---:|
> |Start positions of answer spans|  0.2873 |  0.3056|
> |End positions of answer spans| 0.2796| 0.2091|
>
> We observe that for both the start and end position, a weak correlation is indicated by both metrics between the answer span positions of CBR-MRC's predictions and those of the retrieved cases. This confirms that our model does not overfit to absolute positions seen in the retrievals.
>
> ---
>
> > **Comment:** *For example, in the main results, the model shows performance gain in EM but a lower F1 score. the author explains this by saying "The model focuses on getting the written answer rather than over-fitting to the answer tokens." However, I feel such a case retrieval will lead to more over-fitting.*
>
> EM and F1 are less informative metrics and sometimes misleading since they do not distinguish cases when a model predicts the right answer tokens in the right v/s the wrong contexts. That is, a model may obtain high EM and F1, but never produce the answer tokens with the correct supporting evidence or for the right reasons. Nonetheless, we report EM and F1 (Table 1) due to their common usage in the literature and observe that CBR-MRC based on BERT-base (1) outperforms not only the baseline BERT-base but also SpanBERT-large, and (2) is surpassed only in the evaluation of NewsQA F1. Notably, on the stricter evaluation setting of Span-EM and Span-F1, CBR-MRC consistently outperforms all baselines, demonstrating its accuracy and reliability.
>
> ---
>
> > **Comment:** *Impact of the casebase is unclear. The method utilizes training data as the casebase for retrieval. It is unclear how that casebase influences the performance of the model. And whether the utilization of training data as the casebase would lead to over-fitting?*
>
> **Response:** In our experiments, we find that using the training data as the casebase consistently helps test performance and does *not* cause overfitting. Note that our case retrieval is based on similarities in question *types*, and CBR-MRC is trained to be able to reuse generalizable reasoning patterns in the cases but not their case-specific entity information.
>
> Our choice to use the training data as the casebase in all experiments was made in order to construct a fair evaluation setting, since the baseline methods have access to the full training data as well. For in-domain evaluation on held-out test sets, we show in Table 1 and 2 that CBR-MRC (based on BERT) outperforms the BERT baseline and does not exhibit overfitting to the training data. Further, for few-shot domain adaptation, results in Section 4.3.3 demonstrate that despite using the NQ training data in the casebase, CBR-MRC is able to generalize well to a new domain by simply augmenting the casebase with a few target domain examples without any parameter updates. As for zero-shot domain adaptation, where no data from a new domain is available, we argue that this setting is similarly challenging for both fully-parametric models and CBR-MRC. Building out-of-domain generalization capabilities using our model represents an exciting avenue for future work.
>
> In summary, using the training data as the casebase does not have an adverse effect on the generalization abilities of the proposed CBR-MRC method.

---

### Official Review · Reviewer_2R1W · 2023-08-05

**Soundness:** 4

**Excitement:**

4: Strong: This paper deepens the understanding of some phenomenon or lowers the barriers to an existing research direction.

**Paper Topic And Main Contributions:**

This work introduces a new semi-parametric method for span-based reading comprehension QA where for the test instance, they find best matching K instances from the training data, and use alignment between the representations to predict the answer. Concretely, it works in the following way:

- Make a collection of (passage, entity-masked question, answer) tuples from training data. For the test question, mask the entities and find the top K training tuples by comparing test and train entity-masked questions. Then extract BERT representations for candidate spans from the passage (all spans containing an entity) of the test instance, and find the best matching candidate span from all top K training tuples' passages. The span in the test instance passage that matches the best is the final answer.
- For top K similarity function to work, the authors train use contrastive loss for ranking answer spans higher than non-answer spans.

This approach has 2 nice features:

1. It provides additional interpretability as one can look at the training instances model uses to make the prediction.
2. It provides a way for the model to use new training questions without having the need to retraining on it, like standard RC model pipelines.

Interestingly, this approach also performs much better and shows improved robustness to lexical changes than baselines on NewsQA and NaturalQuestions.

**Reasons To Accept:**

- This work introduces quite a novel approach to RC QA. The ideas beind this work can potentially be applied to other tasks as well.
- The interpretability and no-training extensibility features of this approach are particularly appealing.
- The proposed models gets good results on the datasets even using representations from small model like BERT.
- The paper is well written.

**Reasons To Reject:**

__(main) All the experiments using the proposed work use BERT-base encoder only.__

BERT models are over 5 years old at this point and even among that, the model used BERT-base is a weaker one. Although the demonstrated gains are large, it's not clear if these gains would hold up if one uses the current generation of models. Using at least somewhat newer and larger models like T5 or DeBERTa would have significantly improved my confidence.


__(main) Unclear if the proposed method is truly much better than baselines.__

The proposed method performs much better on EM (8-12) points but is within ~2 points (better or worse) on F1. For span-based evaluation where the same entity can occur multiple times referred to in different ways (President Obama vs Barack Obama), F1 seems a better metric than EM. F1 of course also has flaws though. Given the difference in the result is so large, it would be useful to manually evaluate and compare ~100 predictions or so by different systems.

__(minor) There are many other RC datasets available, which would be easy to apply this method on.__

It would have been better to see the results on a couple of more datasets, at least common ones like Squad.

__(not weakness, suggestion) Some discussion of how this relates to nearest-neighbor models would be helpful.__

Although I'm not familiar with this line of work, and I cannot find paper right now, I think I've seen works where they find the best matching instance for the test instance from the training data and use its classification label for the test instance. I think I've only seen it for classification tasks though.

**Reproducibility:**

5: Could easily reproduce the results.

**Reviewer Confidence:**

4: Quite sure. I tried to check the important points carefully. It's unlikely, though conceivable, that I missed something that should affect my ratings.

---

> ### Author Rebuttal · Authors · 2023-08-29
>
> Thank you for the constructive review! We appreciate that you found our work novel (echoed by paQh and Crp1), well-written (echoed by paQh), interpretable (echoed by paQh and Crp1), and generalizable to other tasks. Please find our detailed responses to your comments and questions below:
>
> > **Comment:** *(main) All the experiments using the proposed work use BERT-base encoder only.*
>
> **Response:** We acknowledge your valid concern and present new results with one of the suggested models (we will be sure to include more models in the camera-ready version). The following are the results of experiments using DeBERTa as the base model for CBR-MRC and for the baseline on the NQ dataset:
>
> | Score | CBR-MRC w/ DeBERTa  | DeBERTa baseline|
> |:---:|:---:|:---:|
> |EM| 0.8030 | 0.4311|
> |F1| 0.8222 | 0.7407|
> |Precision| 0.8625 |  0.8410 |
> |Recall| 0.8254| 0.7182|
> |Span EM| 0.7609| 0.0036|
> |Span F1| 0.8039 | 0.6370 |
> |Span Precision| 0.8450 | 0.8037 |
> |Span Recall| 0.8050| 0.5679|
>
> These results confirm that our method similarly demonstrates strong empirical performance with more recent models. We will add these and a more expanded set of results with a selection of newer models in the final version of the paper. Thank you for raising this concern!
>
> ---
>
> > **Comment:** *(main) Unclear if the proposed method is truly much better than baselines.*
>
> **Response:** We argue that vanilla EM and F1 are less sound metrics for MRC since they do not distinguish between cases when a model predicts the right answer tokens in the right v/s the wrong context. That is, a model may obtain high F1 but never produce answers for the right reasons. We, therefore, report Span-EM and Span-F1 (Table 2), which require a model to predict not only the correct answer tokens, but also their right occurrences in the passages, thus eliminating spurious hits. In our experiments, we find that CBR-MRC consistently outperforms all baselines on both Span-EM and Span-F1 on this stricter evaluation. We do also report EM and F1 (Table 1) due to their common usage in the literature, and observe that CBR-MRC based on BERT-base (1) outperforms not only BERT-base but also SpanBERT-large, and (2) is surpassed only in the evaluation of NewsQA F1.
>
> The following examples illustrate the weakness of the vanilla F1 metric, where the baseline BLANC model is given partial credit despite predicting incorrect spans due to overlapping tokens with the ground truth answer; crucially, Span-F1 is robust to such cases.
>
> ```
> question: where does the term jack mormon come from
> groundtruth: “jackson county , missouri”
> prediction: “missouri , during the kirtland period of latter day saint history , circa 1834”
> ```
>
> ```
> question: which river separates the bronx in new york city from manhattan island
> groundtruth: “harlem river”
> prediction: “the hudson river”
> ```
>
> In summary, on the more interpretable and challenging metrics of Span-EM and Span-F1 (Table 2), CBR-MRC consistently achieves state-of-the-art performance and exhibits both accuracy and better reliability than baseline methods.
>
> ---
>
> > **Comment:** *(minor) There are many other RC datasets available, which would be easy to apply this method on.*
>
> Thank you for the suggestion! Our decision to prioritize the evaluated datasets was motivated by the available room for improvement based on prior art. We, therefore, did not evaluate on datasets such as SQuAD, which are near saturation (>93% EM). However, we will be sure to include a selection of a few more challenging datasets in the camera-ready version.
>
> ---
>
> > **Comment:** *(not weakness, suggestion) Some discussion of how this relates to nearest-neighbor models would be helpful.*
>
> **Response:** Thank you for the helpful suggestion! A canonical work that uses nearest-neighbor retrieval for question answering, probably also the paper being referred to in the review, is PAQ [1]. Here, for a test question, the most similar question from a large set of synthetically generated questions is retrieved and the answer corresponding to the retrieved question is simply output as the prediction. In contrast, our CBR-MRC method differs in the following crucial ways - (a) we retrieve not just one but a set of questions that are jointly reasoned over, (b) the retrieval aims to select questions with the same reasoning patterns but not necessarily the same entities as the test question, and (c) the actual answer tokens of the retrieved questions are not used as the prediction - we instead find the spans in the accompanying context/passage of the test question whose neural representations are the most similar to the representations of the answers of the retrieved questions. Crucially, nearest-neighbor methods, such as PAQ, retrieve questions and output their labels as predictions, and thus perform no reasoning. In contrast, our method uses an explicit reasoning mechanism over the retrieved set of QA pairs.
>
> We will be sure to include this discussion in the final version of the paper. Please let us know in case you had a different work in mind - we would be happy to add more discussion.
>
> ---
>
> [1] Patrick Lewis, Yuxiang Wu, Linqing Liu, Pasquale Minervini, Heinrich Küttler, Aleksandra Piktus, Pontus Stenetorp, and Sebastian Riedel. "PAQ: 65 Million Probably-Asked Questions and What You Can Do With Them." 2021. arXiv preprint arXiv:2102.07033

---

### Official Review · Reviewer_paQh · 2023-08-12

**Soundness:** 4

**Ethical Concerns:**

Yes

**Excitement:**

4: Strong: This paper deepens the understanding of some phenomenon or lowers the barriers to an existing research direction.

**Missing References:**

1. Gao, Tianyu, Adam Fisch, and Danqi Chen. "Making pre-trained language models better few-shot learners." arXiv preprint arXiv:2012.15723 (2020).
2. Lewis, Patrick, Pontus Stenetorp, and Sebastian Riedel. "Question and answer test-train overlap in open-domain question answering datasets." arXiv preprint arXiv:2008.02637 (2020).
3. Wang, Cunxiang, Pai Liu, and Yue Zhang. "Can Generative Pre-trained Language Models Serve As Knowledge Bases for Closed-book QA?." Proceedings of the 59th Annual Meeting of the Association for Computational Linguistics and the 11th International Joint Conference on Natural Language Processing (Volume 1: Long Papers). 2021.
4. Zhang, Hongxin, et al. "Robustness of Demonstration-based Learning Under Limited Data Scenario." Proceedings of the 2022 Conference on Empirical Methods in Natural Language Processing. 2022.

**Paper Topic And Main Contributions:**

This work proposes a novel approach using similar question-answer-context triples in training sets as context to solve machine reading comprehension, which is motivated by similar questions that can attribute to specific predictions. The authors use [cls] of BERT to calculate the similarity of questions. They mask the entities in question to remove spurious similarity caused by entities. Then they rerank the candidate answers according to their similarity scores with retrieved similar answers. The experiments are conducted on four MRC datasets and achieve the SOTA performance. The results also show the effectiveness of the few-shot setting.

**Questions For The Authors:**

How to extract the candidate answer span? Using existing parsing tool?
Have there been any statistics about the accuracy of the question that the answer is in the answer of similar question pairs and not in the answer? If no similar answers exist in the fields used, is this method still very useful?

**Reasons To Accept:**

1. The method is novel in MRC. The authors propose a new way to retrieve questions that can ease spurious similarity and use a more explainable way to rerank retrieved answers instead of directly concatenating them together.
2. The paper is well-organized and easy to read.
3. The method is straightforward and works well in different settings.

**Reasons To Reject:**

I am not very sure about the most essential reason for the success of this method. Whether the golden answer appears in the retrieved answers is what makes the method effective. I think it would be better if more ablation studies can be conducted for a more solid conclusion:
1. Concatenating samples in training sets and original input to predict without reranking answers with similarity scores.
2. Calculate the correlation between prediction accuracy and the answer overlap of retrieved answers and golden answers. I assume if the golden answer does not exist in retrieved answers the performance will decrease heavily.

**Reproducibility:**

4: Could mostly reproduce the results, but there may be some variation because of sample variance or minor variations in their interpretation of the protocol or method.

**Reviewer Confidence:**

3: Pretty sure, but there's a chance I missed something. Although I have a good feel for this area in general, I did not carefully check the paper's details, e.g., the math, experimental design, or novelty.

**Typos Grammar Style And Presentation Improvements:**

Page 7, Line 487: albeit --> ALBERT
Suggest sharing the source code.

---

> ### Author Rebuttal · Authors · 2023-08-29
>
> Thank you for the thoughtful review! We appreciate that you found our work novel (echoed by 2R1W and Crp1), well-written (echoed by 2R1W), applicable in multiple settings, and interpretable (echoed by 2R1W and Crp1). Please find our detailed responses to your comments and questions below:
>
> > **Comment**: *I am not very sure about the most essential reason for the success of this method. Whether the golden answer appears in the retrieved answers is what makes the method effective.*
>
> **Response**: The main hypothesis underlying CBR-MRC is that *answers to similar questions share semantic similarities*. We take a semi-parametric approach to explicitly reason over a set of answer spans of questions similar to the test question, where similarity is computed using contextualized representations from a language model. We, thus, reduce the task of extractive QA to the selection of a span from the test passage that is most similar to the answer spans of similar questions. Crucially, we use *representations* of the retrieved answers to select the answer span for the test question and, therefore, **CBR-MRC does not require the gold answer to exist in the retrieved set**.
>
> ---
>
> > **Comment**: *I think it would be better if more ablation studies can be conducted for a more solid conclusion: 1. Concatenating samples in training sets and original input to predict without reranking answers with similarity scores.*
>
> **Response**: The idea to augment the input with cases has indeed been explored in prior QA works, such as in KBQA [1], semantic parsing [2], and commonsense KBC [3]. While these methods similarly make use of a set of retrievals, the reasoning mechanism they employ is fully parametric and thus black-boxed. Our method takes a substantially different approach to using retrievals by providing an explicit algorithm for span selection that is transparent and interpretable. However, exploring a parametric method of combining retrievals to make predictions for MRC is a very interesting future direction that we could explore. Thank you for the suggestion!
>
> ---
>
> > **Comment**: *2. Calculate the correlation between prediction accuracy and the answer overlap of retrieved answers and golden answers. I assume if the golden answer does not exist in retrieved answers the performance will decrease heavily.*
>
> > **Question**: *Have there been any statistics about the accuracy of the question that the answer is in the answer of similar question pairs and not in the answer? If no similar answers exist in the fields used, is this method still very useful?*
>
> **Response**: We briefly discuss the NQ dataset as evidence for our claim that the gold answer need not be in the retrievals. In NQ, there are 8,061 (~63%) test questions **with answers that are not seen in the training set**. The accuracy of our method on this subset is 71.8% (+16.7% compared to BLANC). In further analysis on CBR-MRC’s retrieve step, we find that only 18 out of 12,836 test questions (0.14%) in fact have the gold answer appearing in the answer set of the retrieved cases. These numbers verify that our method does not need the gold answer to appear within the retrieved cases.
>
> ---
>
> > **Question**: *How to extract the candidate answer span? Using existing parsing tool?*
>
> **Response**: Our approach follows prior QA work [4] and selects a broad set of spans that may be answers to factoid-type questions, such as entity mentions, date-time strings, numbers, and quoted strings. We also include all n-grams up to three words. For date-time string extraction, we use the `datefinder` Python package, and for entity mentions, we use the T-NER library [5].
> We present this discussion in L313-322 of Section 3.2 (“Case Reuse”), and will include these additional details in the camera-ready version.
>
> ---
>
> **Missing References**: Thank you for pointing these out. We will add them in our final version.
>
> ---
>
> [1] Das, R., Zaheer, M., Thai, D., Godbole, A., Perez, E., Lee, J. Y., Tan, L., Polymenakos, L., & McCallum, A. (2021). Case-based Reasoning for Natural Language Queries over Knowledge Bases. In EMNLP 2021
>
> [2] Pasupat, P., Zhang, Y., & Guu, K. (2021). Controllable Semantic Parsing via Retrieval Augmentation. In EMNLP 2021
>
> [3] Yang, Z., Du, X., Cambria, E., & Cardie, C. (2023). End-to-end Case-Based Reasoning for Commonsense Knowledge Base Completion. In EACL 2023
>
> [4] Seo, M., Lee, J., Kwiatkowski, T., Parikh, A. P., Farhadi, A., & Hajishirzi, H. (2019). Real-time open-domain question answering with dense-sparse phrase index. In ACL 2019.
>
> [5] Ushio, Asahi, and Jose Camacho-Collados. "T-NER: An All-Round Python Library for Transformer-based Named Entity Recognition." In EACL 2021

---

### Meta-Review · Senior_Area_Chairs · 2023-09-16

**Recommendation:** 5

**Metareview:**

This paper proposes a case based reasoning approach to reading comprehension question answering. They use BERT embeddings to identify similar questions in the training data to a given test example, and use these to help predict the final answer. The reviewers found the approach presented to be novel for these sorts of QA tasks, and that the paper was well-written. The results were also strong, and during rebuttal the authors provided additional results with DeBERTa that strengthened their contribution.

Unfortunately, the authors missed the following paper, which seems to be a generalization of their work:

Roshni G. Iyer, Thuy Vu, Alessandro Moschitti, and Yizhou Sun. Question-Answer Sentence Graph for Joint Modeling Answer Selection. The 17th Conference of the European Chapter of the Association for Computational Linguistics (EACL 2023)

---

### Decision · Program_Chairs · 2023-10-07

**Decision:**

Accept-Findings

**Comment:**

This paper proposes a case based reasoning approach to reading comprehension question answering. They use BERT embeddings to identify similar questions in the training data to a given test example, and use these to help predict the final answer. The reviewers found the approach presented to be novel for these sorts of QA tasks, and that the paper was well-written. The results were also strong, and during rebuttal the authors provided additional results with DeBERTa that strengthened their contribution.

Unfortunately, the authors missed the following paper, which seems to be a generalization of their work:

Roshni G. Iyer, Thuy Vu, Alessandro Moschitti, and Yizhou Sun. Question-Answer Sentence Graph for Joint Modeling Answer Selection. The 17th Conference of the European Chapter of the Association for Computational Linguistics (EACL 2023)